# Construction of Multifunctional Hierarchical Biofilms for Highly Sensitive and Weather-Resistant Fire Warning

**DOI:** 10.3390/polym15183666

**Published:** 2023-09-06

**Authors:** Tongtong Ma, Qianqian Zhou, Chaozheng Liu, Liping Li, Chuigen Guo, Changtong Mei

**Affiliations:** 1College of Materials Science and Engineering, Nanjing Forestry University, Nanjing 210037, China; matongtongtsmc@163.com (T.M.); emper@njfu.edu.cn (C.L.); 2Key Laboratory for Biobased Materials and Energy of Ministry of Education, Institute of Biomass Engineering, South China Agricultural University, Guangzhou 510642, China

**Keywords:** fire-warning, biofilms, fire resistance, superhydrophobicity, anti-ultraviolet

## Abstract

Multifunctional biofilms with early fire-warning capabilities are highly necessary for various indoor and outdoor applications, but a rational design of intelligent fire alarm films with strong weather resistance remains a major challenge. Herein, a multiscale hierarchical biofilm based on lignocellulose nanofibrils (LCNFs), carbon nanotubes (CNTs) and TiO_2_ was developed through a vacuum-assisted alternate self-assembly and dipping method. Then, an early fire-warning system that changes from an insulating state to a conductive one was designed, relying on the rapid carbonization of LCNFs together with the unique electronic excitation characteristics of TiO_2_. Typically, the L-CNT-TiO_2_ film exhibited an ultrasensitive fire-response signal of ~0.30 s and a long-term warning time of ~1238 s when a fire disaster was about to occur, demonstrating a reliable fire-alarm performance and promising flame-resistance ability. More importantly, the L-CNT-TiO_2_ biofilm also possessed a water contact angle (WCA) of 166 ± 1° and an ultraviolet protection factor (UPF) as high as 2000, resulting in excellent superhydrophobicity, antifouling, self-cleaning as well as incredible anti-ultraviolet (UV) capabilities. This work offers an innovative strategy for developing advanced intelligent films for fire safety and prevention applications, which holds great promise for the field of building materials.

## 1. Introduction

Frequent fire accidents have caused considerable damage to surroundings, human beings, economic and social resources [1,2]. There are classic cases of major fire incidents such as Paris’s Notre Dame Fire, in 2019, Brazil’s National Museum Fire, in 2018, Grenfell Tower in London, in 2017 and the Tianjin Port Explosion, in 2015, etc., [3,4]. Therefore, it is imperative to prevent and alleviate fire hazards to improve fire safety and protect people’s lives and property [5]. At present, endowing combustible materials with flame retardancy together with constructing early fire-warning devices are two powerful ways to avoid fire incidents. Compared to the former, fire-warning strategies allow more time for fire-fighting and rescue by being able to respond before a fire happens [6,7]. Up to now, gas, smoke and infrared heat detectors have been widely used in commercial applications, showing good detection capabilities in closed, indoor environments [5,8]. However, there are still some fatal drawbacks in terms of the delayed warning, long response time (>100 s) and their very limited application in complex outdoor surroundings, especially in extreme weather such as heavy rain, strong solar storms, or corrosive environments [9,10]. Hence, it is imperative, but also challenging, to construct an early-warning fire-alarm device with extremely high sensitivity and ultrafast response, that is flame retardant and gives a weather-resistant performance to provide fire safety for combustible materials.

Recently, employing temperature-sensitive materials to develop early fire-warning detectors to monitor high-temperature heat sources was investigated, offering an innovative strategy for reducing the fire hazard of inherently flammable materials [11]. For example, various kinds of fire-warning sensors made of graphene oxide (GO) or MXene were reported with rapid fire-response behaviors [7,12]. However, the flame-retardant capabilities of these carbon-based materials were limited, and long-term exposure to fire would cause circuit failure, which, used alone, could not meet the needs of fire safety. Chen et al. [13] prepared a GO/carbon nanotubes (CNTs) hybrid multilayered coating on wood pulp paper (WPP) via a universal layer-by-layer (LBL) assembly, which was used as an ideal fire alarm sensor and responded quickly after exposure to flame for 5 s. The corresponding response time can be adjusted by changing the content of the CNTs in the composite conductive network. Notably, CNT is another promising carbon nanomaterial that can be regarded as a graphene cylinder rolled at a certain angle, which shares numerous similar characteristics with graphene, containing desirable chemical and heat endurance, as well as rich electronic and optical properties [14]. These particular advantages render CNT a reliable sensing component for constructing high-performing fire-warning systems [13]. Generally, the conductive CNTs were modified with organic molecules such as chitosan (CS), polyvinyl alcohol (PVA) or by establishing insulating layers such as cellulose on their surfaces to achieve the initial insulating states [15,16]. Further, these grafted molecules or constructed layers would form the conductive char layers when exposed to a high temperature or fire, thereby enabling a rapid transition in resistance and sending an alarm signal. Xia. et al. [15] manufactured a fire-warning sensor by coating carbon black nanoparticles, PVA and CNT on cotton fabric, thus exhibiting a shorter alarm time of ~4 s under fire. Chen. et al. [17] built a CS-based CNT-hybrid coating on polyurethane sponge (PUS) using a layer-by-layer method, which can be employed as a fire-warning sensor with a trigger time of ~1 s. Nevertheless, these biomolecular materials exhibited strong hygroscopic and poor ultraviolet (UV)-light stability owing to the presence of abundant polar groups such as OH and COOH in the chains [18]. When combined with CNTs to prepare early-warning sensors, it is very easy to cause irreversible reactions such as oxidation, degradation, and cross-linking, which greatly limits the durability for long-term use in extreme outdoor environments such as heavy rain, strong solar storms and corrosive environments.

Nanoscale materials have been widely used in the preparation of superhydrophobic and UV-resistant surfaces because of their special physical and chemical characteristics [19]. As an important photofunctional material, TiO_2_ simultaneously expresses the advantages of high refraction index, a UV-ray shielding property, biological inertness, great thermostability and high security, which are important when fabricating a multifunctional surface [20,21]. Importantly, intrinsically hydrophilic TiO_2_ is easily modified with low-surface-energy materials such as trimethoxy(octadecyl)silane (TMODS), fluoropolymer, hexadecyltrimethoxysilane (HDTMS), giving it the potential to become a superhydrophobic material [22,23]. In this regard, hydrophobic TiO_2_ nanoparticles have been extensively applied on the surface of cellulose papers and cotton fibers, which simultaneously endow the material with superhydrophobic, UV-resistant, and self-cleaning properties [24,25]. Therefore, it makes sense to integrate TiO_2_ nanoparticles into the surface of a CNT-based early fire-warning system through its structural design, which would not only further improve its thermal stability and fire prevention ability, but also endow it with waterproof, anti-UV-aging and self-cleaning properties, thus providing a new idea for the preparation of an early-warning sensor with good comprehensive performance.

Herein, a multifunctional hierarchical biofilm with significant flame retardancy, superhydrophobicity and UV resistance was fabricated using CNTs, LCNF and TiO_2_ building blocks through an ecofriendly and flexible method, which can serve as an ultrasensitive fire sensor and a self-cutting fuse, thus achieving early-fire warning and guaranteeing circuit safety synchronously. Firstly, as exhibited in Figure 1a, a three-layer “sandwich-like” film consisting of upper, middle and bottom layers was constructed by a vacuum-assisted alternate self-assembly method. Both the upper and lower layers are made of LCNF and cover the surface of the CNT layer as a protective layer to give it a low-temperature, insulating property; the middle layer is assembled from CNTs, emphasizing its function as a fire sensor for detecting temperature variation or fire attack. Next, the “sandwich-like” film was modified with fluorine-free HDTMS to obtain the hydrophobic TiO_2_-HDTMS layer by dip-coating, thus resulting in a multiscale hierarchical fire-warning film. Further, Figure 1b shows that the fire-warning device was assembled by linking the hierarchical film with a low-voltage power supply and an alarm light via many electric wires. When encountering a fire attack, the upper- and bottom-insulating LCNF layers can be rapidly carbonized at a high temperature, turning the insulated hierarchical film into a conductive one; meanwhile, the unique electron excitation of the TiO_2_ is thus activated to cause a sensitive resistance transition, thereby giving a timely early fire-alarm signal at ~0.30 s. The structure and topography, early fire-warning performance, flame retardant, hydrophobic and UV-resistant properties of the hierarchical films were systematically investigated. Importantly, the hierarchical-structure design enables the biofilm to simultaneously possess excellent fire resistance to alcohol lamps for ~1238 s, a high water contact angle of 166 ± 1°, and an outstanding ultraviolet protection factor (UPF) value of 2000. Therefore, our work may pave an innovative way for the design and construction of a multifunctional hierarchical biofilm as an ultrasensitive fire-warning sensor, with potential applications in construction, and indoor and outdoor decorations.

## 2. Experimental Section

### 2.1. Materials

CNTs were purchased from Tanfeng Technology Co., Ltd., (Suzhou, China). Pulp was supplied by Ruifeng Paper Co., Ltd., (Jiaozuo, China). Nano-TiO_2_ particles (~15 nm in diameter), 2,2,6,6-tetramethylepiperidin-1-oxyl (TEMPO) and trimethoxyhexadecylsilane (HDTMS, CH_3_(CH_2_)_15_Si(OCH_3_)_3_) were obtained from Macklin Biochemical Co., Ltd., (Shanghai, China). Ethanol, sodium hydroxide (NaOH) and sodium bromide (NaBr) were purchased from Aladdin Bio-Chem Technology Co., Ltd., (Shanghai, China). Ultra-pure deionized water was prepared from a reverse osmosis water system in our laboratory.

### 2.2. Measurements

The microstructures of the LCNF and CNT were observed by a JEM-1400 transmission electron microscopy (TEM, JEOL Co., Tokyo, Japan) and Tecnai 12 TEM (Philips, Amsterdam, The Netherlands) under the acceleration voltage of 15 kV and 200 kV, respectively. X-ray diffraction (XRD) patterns were recorded by a XRD Ultima IV diffractometer (Rigaku, Tokyo, Japan), using CuKα radiation operated at 20 kV and 20 mA, over a 2θ range of 5 to 70° at a scanning rate of 10°·min^−1^. The chemical constitution analysis of the films was analyzed using an AXIS-ultra DLD X-ray photoelectron spectrometer (XPS, Kratos, Manchester, UK), employing focused monochromatized Al Kα radiation (hν = 1486.6 eV) at a power of 600 W. The morphology and microstructure of the film as well as the corresponding char layers after burning were observed using a Regulus 8100 scanning electron microscope (SEM, Hitachi, Tokyo, Japan) equipped with an Ultima Max 170 X-ray energy dispersive spectroscope (EDS, Oxford Instrument, Oxford, UK) at an accelerating voltage of 3.0 kV. The thermo-gravimetric analysis (TGA) of the films was performed on a TG 209 F1 instrument (Netzsch, Waldkraiburg, Germany) using a heating rate of 10 °C·min^−1^ from 30 to 800 °C under air atmosphere. The water contact angles (WCAs) of the film surfaces before and after TiO_2_-HDTMS modification were characterized by DSA 100 m (KRUSS, Hamburg, Germany) using a sessile drop technique. The values of the WCAs were obtained by testing five different locations and calculating the average values. UV-resistance performances of the films were evaluated by UV-2000 ultraviolet transmittance tester (Labsphere, North Sutton, NH, USA) in the wavelength of 290–400 nm.

### 2.3. Preparation of LCNFs

Firstly, 20 g of pulp was washed in 70 °C water and 50 °C ethanol for 3 h, and then dried to obtain the pretreated pulp sample. Next, 5 g of pretreated pulp was suspended in 500 mL aqueous solution including 0.016 g of TEMPO and 0.10 g of NaBr. Then, TEMPO-mediated oxidation was carried out by adding 6 g of 10 M NaClO solution dropwise to the above solution and stirring continuously at room temperature for 2 h. Meanwhile, the pH was maintained around 10 by the addition of 0.5 M NaOH solution. After the reaction, the resulting mixture was washed with distilled water to remove the residual chemicals until pH was neutral. Finally, the washed pulp was diluted to a 5 M suspension and then ultrasonically treated under conditions of 1500 W output power for 30 min using an ultrasonic cell crusher, thus resulting in an LCNF dispersion liquid.

### 2.4. Preparation of the Hierarchical Films

The multifunctional hierarchical films were fabricated by using CNTs, LCNF and TiO_2_ building blocks by alternate self-assembly together with a dip-coating method. As shown in Figure 1a, 4 mL dispersion liquid of LCNF with a concentration of 5 mg·mL^−1^ was first poured into a sand core filter funnel with a PVDF microporous film (pore size 0.45 μm); after vacuum filtration and drying, 6 mL suspension of CNTs with a concentration of 6 mg·mL^−1^ was poured onto the LCNF layer and filtrated to remove water; then another 4 mL of LCNF dispersion liquid was poured on the CNT layer and dried by vacuum filtration, thus resulting in a sandwich-like film with the name of L-CNT. Afterward, the “sandwich-like” film was modified with the fluorine-free HDTMS chemical to obtain the hydrophobic TiO_2_-HDTMS layers by dip-coating, which includes steps as follows: 1.0 g TiO_2_ particles and 10 μL of HDTMS were dispersed in 50 mL of ethanol under magnetic stirring at 45 °C for 4 h to prepare the dip-solution; the “sandwich-like” film was then immersed in the above solution and heated to 60 °C for 2 h to fabricate TiO_2_-HDTMS layers. Finally, a multiscale hierarchical fire-warning film was obtained after drying at 40 °C, and was recorded as L-CNT-TiO_2_.

## 3. Results and Discussion

### 3.1. Structure and Thermal Stability Characterizations

The morphologies and diameter distributions of the LCNFs and CNTs were investigated by TEM. As shown in the images of the negatively stained LCNF preparation in Figure 2(a_1_,a_2_), all the noodle-like LCNFs possess a mean diameter of ~10  nm, which can form a highly entangled network. Furthermore, it was found from Figure 2(a_2_) that some irregular aggregate co-appeared along with the noodle-like LCNFs. These aggregates may be composed of lignin, which are consistent with those reported in the literature [26], thus indicating the successful production of LCNFs. In addition, the entangled tubular-CNT network with hollow pores and a multilayer-wall structure can be observed in Figure 2(b_1_,b_2_). As shown, the outer and inner diameters of the CNTs were 8–10 nm and 14–16 nm, respectively. Furthermore, there were approximately 20 tube walls in each tube and the wall interlayer spacing was about 0.34 nm, which laid the foundation for its excellent electrical conductivity.

The surface chemical compositions of the L-CNT and L-CNT-TiO_2_ films were characterized by XPS, and the corresponding results are shown in Figure 2(c_1_–c_3_,d_1_–d_3_). From Figure 2(c_1_), the XPS spectra of the L-CNT film only presented two typical peaks of C 1s (284.80 eV) and O 1s (530.32 eV) with contents of 74.61 and 25.39%, respectively. However, after TiO_2_-HDTMS modification, Figure 2(d_1_) shows that two new peaks of Ti 2p (458.98 eV) and Si 2p (102.18 eV) with contents of 11.74 and 9.67% appeared in the survey scan spectrum of the L-CNT-TiO_2_ film. The results demonstrated the successful preparation of the TiO_2_ layer on the surface of L-CNT film. Notably, the Ti and Si elements originating from the TiO_2_ and HDTMS molecules that can play active roles in improving both flame retardancy and hydrophobic properties of the film. Further, the corresponding high-resolution XPS curves of the C 1s with peak-fitting spectra of the L-CNT and L-CNT-TiO_2_ films are displayed in Figure 2(c_2_,d_2_). As exhibited, both the films had three distinct characteristic peaks at 284.80 eV, 285.33 eV, and 286.51 eV, corresponding to C-C/C-H, C-O and C=O/O-C-O bonding states, respectively. Comparatively, the contents of C-O and C=O/O-C-O bonding states in L-CNT-TiO_2_ film were higher than those in L-CNT film, which was attributed to the in situ packing of TiO_2_ nanoparticles on the LCNF layers. This phenomenon was also proved by the corresponding O 1s spectra. As shown in Figure 2(c_3_,d_3_), except for the O-C (532.59/532.44 eV) and O=C (532.07/532.07 eV) bonding states that were derived from LCNF molecular chains, a new O-Ti (530.44 eV) state also appeared in the L-CNT-TiO_2_ film, which originated from the hydrophobic TiO_2_ layer on the surface of the L-CNT-TiO_2_ film. These XPS results indicated that both L-CNT and L-CNT-TiO_2_ films had been successfully prepared.

The crystal structural variations of LCNFs, CNTs, TiO_2_ and the resulting L-CNT as well as L-CNT-TiO_2_ films were further characterized by XRD patterns. As shown in Figure 2e, the LCNFs exhibited two crystal planes (1 1 0 and 0 0 2) at 15.05 and 22.46°, respectively. Furthermore, the XRD pattern of CNTs showed a sharp diffraction peak at 26.02° together with a weak peak at 43.06°. As a result, for the L-CNT film, these four diffraction peaks also appeared at 15.40, 22.68, 26.08 and 42.93°, which corresponded well with the (1 1 0) and (0 0 2) planes for LCNFs as well as the (0 0 2) and (1 0 0) planes for CNTs, suggesting the successful assembly of the L-CNT film by LCNFs and CNTs. Comparatively, for L-CNT-TiO_2_ film, except for the four crystal planes mentioned above, an obvious crystal plane (1 0 1) of TiO_2_ also appeared at 25.92°, in Figure 2e. The results demonstrated that the L-CNT-TiO_2_ hierarchical film was successfully fabricated via the further dipping method, which was in accordance with the analysis of the XPS results.

Thermal stability is an important indicator to measure the high-efficiency flame-retardant performance of materials when exposed to fire. The thermal stability of the L-CNT and L-CNT-TiO_2_ films was studied by TG technology under air conditions, and the related results are revealed in Figure 2(f_1_,f_2_). It can be observed from Figure 2(f_1_,f_2_) that the degradation behaviors of the films were divided into three steps with the increase in temperature. The first step of degradation at around 100 °C was mainly ascribed to the evaporation of the stored water in the π-stacked structure [27]. The second step of 300–400 °C was attributed to the degradation of the unstable oxygen-containing lignocellulose molecular chains and the heat removal of the oxidizing groups [28]; The third step of degradation occured at 400–500 °C, which was due to the degradation of the CNT molecular chains and the transformation of the crystal types [29]. Notably, as shown in Figure 2(f_2_), the L-CNT-TiO_2_ presented a maximum thermal degradation rate temperature (T_max_) of 340.6 °C, which was 9.0 °C higher than that of the L-CNT film. The enhancement of T_max_ after covering the TiO_2_ layer was mainly due to the blocking of oxygen and thermal energy, thus delaying the further thermal degradation of the LCNF layers. As a result, the char residue of the L-CNT-TiO_2_ film was significantly improved from 68.54 wt% for L-CNT to 71.10 wt% at 800 °C, demonstrating a higher thermal stability after TiO_2_-HDTMS modification, which provides a reliable foundation for realizing a long-term fire-warning response.

### 3.2. Surface and Cross-Section Morphology Observations

The surface and cross-section morphologies have an important impact on the wettability and fire resistance of the films. The digital photographs of the L-CNT and L-CNT-TiO_2_ films are displayed in Figure 3(a_1_,b_1_). As shown in Figure 3(a_1_), the L-CNT film presented a black and smooth surface morphology because of the alternate self-assembly method. However, after TiO_2_-HDTMS modification, it can be seen from Figure 3(b_1_) that a white TiO_2_ layer was uniformly distributed on the surface of L-CNT-TiO_2_ film. Further, the corresponding SEM surface morphologies of the L-CNT and L-CNT-TiO_2_ are shown in Figure 3(a_2_–a_4_,b_2_–b_4_). Figure 3(a_2_–a_4_) present a relatively flat surface with more irregular streaks, which was attributed to the CNTs entanglement between LCNF layers in the L-CNT film. Comparatively, a very rough surface with more agglomeration TiO_2_ particles is exhibited in Figure 3(b_2_). Furthermore, it can be seen from the corresponding high-resolution SEM images in Figure 3(b_3_,b_4_) that many micro-papillae and nanoscale protrusions were distributed on the surface of the hierarchical L-CNT-TiO_2_ film. The air in the groove below the liquid greatly decreased the contact area between the surfaces of the solid and liquid elements [30]. Overall, both the rough hierarchical structure of the agglomerated TiO_2_ nanoparticles and spaces between them contributed to the surface trapping air and repulsing water, which provided a justification for the preparation of superhydrophobic layers on the surface of the L-CNT-TiO_2_ film.

Moreover, Figure 4 present the images of the cross-section SEM morphologies as well as the EDS element distributions of the L-CNT and L-CNT-TiO_2_ films. For the L-CNT film, as can be seen from Figure 4(a_1_), a typical sandwich-like microstructure with a thickness of 82.0 μm was fabricated by the insulated LCNF layers and the well-conducting CNTs layers sandwiched between the LCNF layers. It is worth noting that a biomimetic nacre-like structure can be observed from the LCNF layer image in Figure 4(a_2_,a_3_), which was formed by closely arranged noodle-like fibers through hydrogen bonds. The conductive interlayer was arranged by CNTs to form a thicker network structure, which provided an important guarantee for maintaining long-term conductivity in fire. However, after TiO_2_-HDTMS modification, Figure 4(b_1_) exhibits a hierarchical microstructure with a thickness of 89.5 μm due to the uniform deposition of the TiO_2_ layers. Moreover, it can be seen from Figure 4(b_2_,b_3_) that a very thin and rough TiO_2_ layer consisting of agglomerated micron and nanoparticles was formed on the surface of the LCNF layer, which is the key to transforming the hydrophilic surface of the L-CNT film into a hydrophobic surface of the L-CNT-TiO_2_. In particular, it can be observed from Figure 4(a_4_) that the interfacial bonding between the insulated LCNF layer and the conductive CNT layer was poor and a clear gap appeared in the interface of the sandwich-like L-CNT film. The existence of such a gap at the interface can promote the rapid detachment of the LCNF layer after charring and thus lose the protective effect on the CNT interlayer in the presence of fire, which has a negative impact on the flame resistance of the L-CNT film and its application in fire warning. Comparatively, when the L-CNT film was modified with the TiO_2_-HDTMS, the gap between the LCNF and CNT layers basically disappeared and was replaced by a tightly bonded inter-laminar structure, as shown in Figure 4(b_4_). The phenomenon was attributed to the coupling effect of methoxy and silanol groups produced by the hydrolysis of HDTMS between the LCNF and CNTs layers, which greatly promoted the interfacial bonding through hydrogen bonding and van der Waals forces.

Furthermore, the corresponding element distribution mappings and compositions of the L-CNT and L-CNT-TiO_2_ films were characterized by EDS, and the results are displayed in Figure 4(c_1_–c_3_,d_1_–d_5_). In Figure 4(c_1_–c_3_), the L-CNT film only contained C and O elements with a content of 67.12 and 32.88%, in which the C element mainly came from the CNT interlayer, and the O element was derived from the LCNF layer. In contrast, as exhibited in Figure 4(d_1_–d_5_), except for the C and O elements, the Ti and Si elements also appeared in the L-CNT-TiO_2_ film, whose contents were 2.21 and 0.24%, respectively. Figure 4(d_4_) reveals that the Ti element signal was evenly dispersed in the upper LCNF layer, demonstrating the successful manufacture of L-CNT-TiO_2_ film. Furthermore, it is worth noting that the Si element signal was well distributed in the entire cross-section of the L-CNT-TiO_2_ film, as seen in Figure 4(d_5_). The appearance of Si-containing groups such as silane can play a coupling role at the interface between layers. Importantly, Si is classified as a flame-retardant element, which can efficiently exert the fire resistance and early-warning capabilities of the resulting film.

### 3.3. Early Fire-Warning and Flame Resistance Behaviors

The early fire-warning behaviors of the films were simulated and monitored via a series circuit consisting of a low-tension power supply, an alarm lamp and several wires, where L-CNT and L-CNT-TiO_2_ films with the size of 20 × 10 mm^2^ acted as the warning sensors. The related fire-warning processes are presented in Appendix A and Figure 5a,b; the relevant data came from three sets of the repeated tests. As can be seen from Appendix A and Figure 5a, the alarm signal of the L-CNT film was triggered after being attacked by the fire for ~0.70 s, which was the result of the transformation of the insulating LCNF layer into a conductive char layer in the presence of fire. As the test progressed, Figure 5a shows that the edges of the film were visibly burned when in contact with the fire for ~272 s. Eventually, the middle of L-CNT film was burnt out after being continuously exposed to the fire for ~544 s, and the circuit changed from the conductive loop to an insulating state, thereby causing the disappearance of the warning signal. Comparatively, as recorded in Appendix A and Figure 5b, the fire-warning trigger time of the L-CNT-TiO_2_ film was reduced from ~0.70 s for L-CNT film to ~0.30 s, which is much shorter than conventional smoke sensors (~100 s) and those reported sensors in the literature (see Table 1). Early-warning signals issued in time before a fire occurs can provide more time for personnel evacuation and fire rescue. Importantly, as exhibited in Figure 1b,c, under continuous flame attack, except for the above carbonization of the LCNF layer, the alarm signal of L-CNT-TiO_2_ film was triggered because the fact that TiO_2_ can generate the thermo-excited electrons and holes owing to the oxygen vacancy-dependent band structure with a low band-gap energy of about 3.2 eV, which can trigger its electron excitation from the valance band (VB) to the conduction band (CB), thereby generating the obvious resistance transition from the insulated state to the conductive one [9]. Further, after TiO_2_-HDTMS modification, it can be observed from Appendix A and Figure 5b that the fire duration time of the L-CNT-TiO_2_ film was improved from ~544 s for L-CNT film to ~1238 s, which was attributed to the protective effect of the thermostable TiO_2_ layer as well as the silicon-containing cross-linked char layer on the conductive CNT interlayer. Generally, the fire duration time of ~1238 s was also longer than those previously reported in the literature (see Table 1), thus suggesting the excellent flame-retardant effect of the constructed TiO_2_ layer and the resulting reliable fire-warning behavior of the L-CNT-TiO_2_ film.

The flame-retardant behaviors of the films before and after TiO_2_-HDTMS modification were visually tested by using a butane spray gun. The corresponding fire temperatures of the tests were 1200–1400 °C, which was equal to the temperature that CNTs can withstand in air. After two consecutive exposures to the fire for 120 s, as can be seen in Appendix A and Figure 5c, the upper- and bottom-LCNF layers of the L-CNT film were completely burned, while the edges of the middle-CNT layer were also partially burned away, which was attributed to the transformation of CNTs into gases such as carbon dioxide during the combustion process. However, for L-CNT-TiO_2_ film, the overall structure of the middle-CNT layer was not burned out by the fire, only the TiO_2_ surface layer was partially destroyed. The results revealed that the fire resistance performance of the TiO_2_-HDTMS modified film were significantly improved. This is because both Ti and Si are flame-retardant elements. When encountering a fire attack, on the one hand, a silicon-containing cross-linked char layer was formed through the interaction of HDTMS with the LCNF molecular chains, which can further combine with the TiO_2_ layer as thermal protection barriers, thereby blocking the heat exchange and oxygen feedback, decreasing the temperature of the burning zone and preventing the spread of the fire; and on the other, the carbonization of the HDTMS distributed in the CNT layer enhanced the char-layer thickness of the CNTs in the middle layer, which can synergistically play the role of efficient flame retardant and fire prevention.

### 3.4. Residual Char Structures and Enhanced Mechanism

The study of the morphologies and compositions of the char layers after combustion is of great significance for the analysis of the flame retardancy-enhanced mechanism of the films. SEM and EDS were employed to analyze the surface morphologies and element distributions of the char layers after being combusted with a butane gun for 60 s, and the resulting images are displayed in Figure 6. It can be seen from Figure 6(a_1_) that the LCNF layer on the surface of the L-CNT film was severely burned, only leaving a small part of the residual char layer. This structure left the conductive CNT interlayer fully exposed to the flame and so could not play a role in flame retardancy. As a result, the CNT interlayer presented an extremely thin and loose morphology with many holes under the high-magnification SEM images, shown in Figure 6(a_2_,a_3_); moreover, a large number of the burned CNTs were sparsely distributed, as shown in Figure 6(a_4_). However, after the TiO_2_-HDTMS modification, a relatively complete and compact protective char layer was uniformly formed on the surface of the L-CNT-TiO_2_ films shown in Figure 6(b_1_,b_2_), which played a crucial role in protecting the CNT interlayer on fire. It is worth noting that a mass of micro- and nano-particles can be visually observed in Figure 6(b_3_,b_4_), which indicated that the construction of the TiO_2_-HDTMS layer enhanced the char-layer strength of the LCNF layer on the surface of the film. This type of silicon-containing char layer is conducive to blocking the exchange of heat and oxygen between the external environment and the CNT interlayer during combustion, thus effectively playing the flame-retardant role and prolonging the fire-warning time. Further, the corresponding char layer components and distribution mappings of the L-CNT film before and after TiO_2_-HDTMS modification are recorded in Figure 6(c_1_–c_4_,d_1_–d_6_). As shown in Figure 6(c_1_–c_4_), only C and O elements appeared on the char layer of the L-CNT film, and the nC/nO value was up to 784.17%, indicating that the LCNF layer was basically burned out. For comparison, for the L-CNT-TiO_2_ film, except for the C and O elements, the Ti and Si elements with contents of 43.24 and 3.18% are also presented in Figure 6(d_5_,d_6_), which originated from the unburned TiO_2_ layer as well as the Si-containing compounds. Furthermore, the nC/nO value of L-CNT-TiO_2_ film was 11.30%, which was 98.56% lower than that of L-CNT film. The results suggested that the dense Si-containing char layer based on TiO_2_ had great thermal stability and fire-resistant ability, which can act as a physical barrier to block the intrusion of heat and flame into the middle conductive-CNT layer, thereby exerting effective flame retardancy and reliable fire-warning behavior.

The elemental compositions of the char layers of the films after combustion was further explored by XPS. As exhibited in the survey XPS spectra in Figure 7(a_1_), the char residue of the L-CNT film only consisted of C and O elements, with the contents of 96.68 and 3.32%, respectively. In contrast, both Ti and Si elements also appeared in the char layer of the L-CNT-TiO_2_ film, shown in Figure 7(b_1_), which were mainly derived from the unburned TiO_2_-protective layer as well as the remaining residue after the pyrolysis and crosslinking of the HDTMS. The results were consistent with those of SEM-EDS analysis. Further, Figure 7(a_2_,a_3_,b_2_,b_3_) display the high-resolution scans of the C 1 s and O 1s graphs for L-CNT and L-CNT-TiO_2_ films. As shown, the C 1 s spectra of both films were divided into four peaks at about 284.80 eV, 285.14/285.24 eV, 286.49/286.33 eV and 289.64 eV, which were attributed to C-C/C-H, C-O, C=O/O-C-O and O-C=O bonding states, respectively. Whereas, for the L-CNT-TiO_2_ film, Figure 7(b_2_) presents the reduced C-C/C-H and improved C-O, C=O/O-C-O and O-C=O bonding states compared with those of the L-CNT film in the char layers, and the resulting nO/nC ratio of L-CNT-TiO_2_ improved from 3.43% for L-CNT film to 13.44%. The results proved that the TiO_2_ layer was not completely destroyed by the fire, while exhibiting good thermal stability and reliable flame-retardant properties. This phenomenon is also proved by the results of the O 1s spectrum analysis. Specifically, it can be observed from Figure 7(b_3_) that O-Ti bonding states appeared in the O 1s spectrum of the L-CNT-TiO_2_ film, which was different from the only O=C and O-C states in the char layers of the L-CNT film (Figure 7(a_3_)). In summary, the well-designed TiO_2_ layer together with the uniformly dispersed Si elements of the L-CNT-TiO_2_ film can protect the CNT interlayer by creating thermostable char layers as a physical barrier against fire, which is of great significance for effectively reducing fire hazard.

### 3.5. Characterization of Superhydrophobic and UV Resistance

The contact angle, anti-adhesion attraction, dynamic anti-wetting behaviors, self-cleaning performances of the films before and after the TiO_2_-HDTMS modification were investigated by water contact angle (WCA) measurement and video recorder, and the results were recorded in Appendix A and Figure 8. As shown in Figure 8(a_1_), the surface of the L-CNT film was exceptionally smooth and hydrophilic, resulting in a low WCA value of 39.5 ± 1°. However, after modification with TiO_2_-HDTMS, the WCA of the L-CNT-TiO_2_ film was improved to 166 ± 1° and still remained unchanged after a few hours, suggesting a perfect transition from a hydrophilic to a superhydrophobic surface together with a reliable wettability resistance. Further, the water drops were carried by a syringe needle and then placed on the surface of the films to monitor their anti-adhesion attraction to each other. As recorded in Appendix A and Figure 8(b_1_), for the L-CNT film, a water droplet was quickly absorbed by the film once it came into contact with the surface, exhibiting poor anti-adhesion properties, while for the L-CNT-TiO_2_ film, once the needle was lifted or moved, the droplet fully left the surface without any remaining (Appendix A and Figure 8(b_2_)), implying an almost non-existent adhesive force on the surface for water. Notably, a water droplet impinging on a surface that is highly hydrophobic and with low-adhesion readily bounces off and escapes from the surface rather than becoming trapped [54]. As shown in Appendix A and Figure 8c, when dropping a water droplet (~6 μL, fall height of 7 cm) on the L-CNT-TiO_2_ film, the droplet could bounce several times off the surface without the surface becoming wet or broken, until the droplet was ejected from the film. The reason is that the surface of L-CNT-TiO_2_ is characterized by the existence of a triple-phase interface; the contact area between the droplet and the TiO_2_ layer is reduced, and the droplet can readily rebound due to the existence of the trapped air. Notably, outstanding self-cleaning ability is a necessary indicator of superhydrophobic materials in practical applications. Appendix A and Figure 8(d_1_,d_2_) record the test processes of the water droplets falling and washing away dust on the L-CNT and L-CNT-TiO_2_ films. For the L-CNT film, as recorded in Appendix A and Figure 8(d_1_), when the water was dropped into the surface made of an LCNF layer, the droplet stayed on its surface firstly due to strong adhesion, and then just moved slowly under gravity without carrying away dust. However, for L-CNT-TiO_2_ films, Appendix A and Figure 8(d_2_) show that the water droplets immediately roll down in spherical form, following the slight inclination of the film and effortlessly carry away dust without leaving a visible trace on the surface, thus demonstrating an unexceptionable self-cleaning capability. This is due to the fact that the interfacial force between the superhydrophobic surface and dust was lower than that between the polar water droplets with dust. In addition, both the L-CNT and L-CNT-TiO_2_ films have a quite low-UV transmittance of 0.05% in the wavelength range of 290–400 nm, which is attributed to the special structure of CNTs with strong UV-absorption capacity. Furthermore, the ultraviolet protection factor (UPF) values of both films are as high as 2000, indicating that they have excellent anti-ultraviolet properties, which can prevent the hierarchical L-CNT-TiO_2_ film from being damaged by ultraviolet radiation, which is of great significance for its practical application outdoors.

## 4. Conclusions

The effective design of intelligent fire-warning biofilms with strong weather resistance plays a vital role in prolonging service life and expanding application fields. In summary, we reported here on a multifunctional hierarchical biofilm (L-CNT-TiO_2_) with an early fire-warning function composed of LCNFs, CNTs and TiO_2_ produced by vacuum-assisted alternate self-assembly and dipping methods. This L-CNT-TiO_2_ film not only exhibited an ultrafast fire-response signal and excellent flame retardancy, but also possessed superhydrophobic, anti-wetting, self-cleaning and UV-resistant properties. Specifically, the insulating film can be translated into a conductive one after being attacked by a fire, thus providing an ultrasensitive fire response signal of ~0.30 s and a ideal alarm time of ~1238 s, which would buy more time for evacuation and fire rescue before the fire breaks out. Furthermore, the L-CNT-TiO_2_ film still basically maintained its original shape even after being burned with a butane spray gun for 120 s, thus contributing to the heat-insulation and flame-retardant effect of the formed dense Si-containing nano-titanium protective layer. Significantly, the L-CNT-TiO_2_ film also possessed a water contact angle (WCA) of 166 ± 1° and an ultraviolet protection factor (UPF) as high as 2000. Overall, this work provides a new route for the development of multifunctional films with an intelligent fire alarm response for potential fire safety and prevention in diverse applications. 

## Figures and Tables

**Figure 1 polymers-15-03666-f001:**
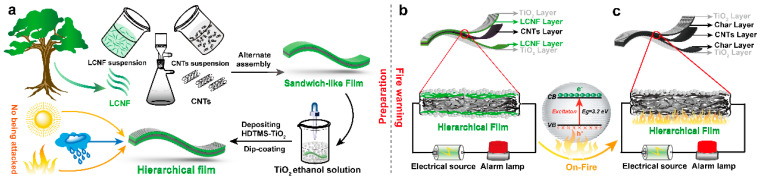
Schematic fabrication process of the hierarchical film L-CNT-TiO_2_ with significant flame retardancy, superhydrophobicity and UV resistance (**a**); experimental circuit diagram of the fire-warning behavior and relevant alarm mechanism (**b**,**c**).

**Figure 2 polymers-15-03666-f002:**
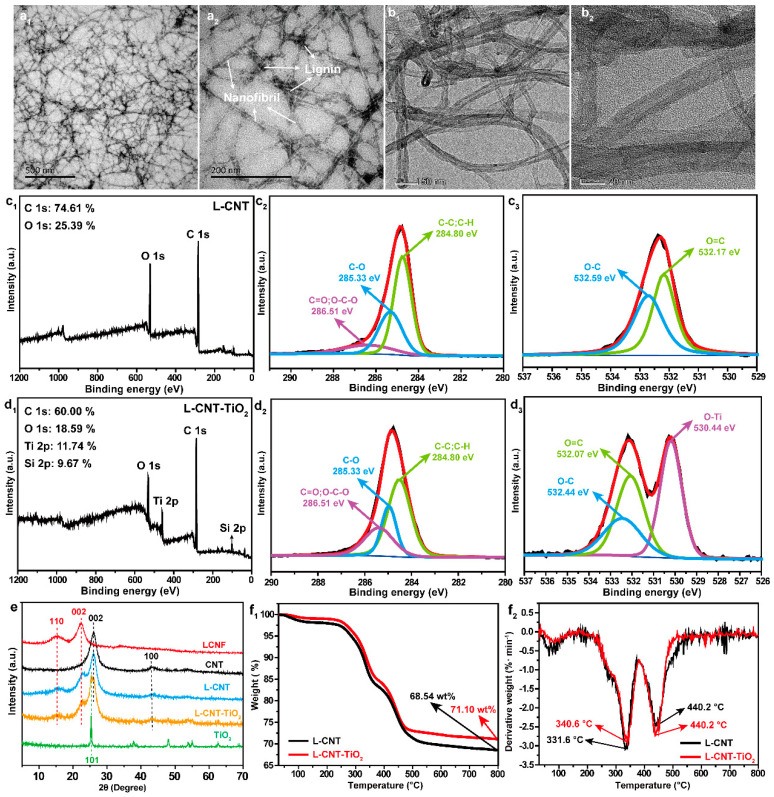
TEM images of LCNFs (**a_1_**,**a_2_**) and CNTs (**b_1_**,**b_2_**); XPS spectra (**c_1_**–**c_3_**,**d_1_**–**d_3_**); XRD patterns (**e**), TG (**f_1_**) and DTG (**f_2_**) curves of the L-CNT and L-CNT-TiO_2_ films.

**Figure 3 polymers-15-03666-f003:**
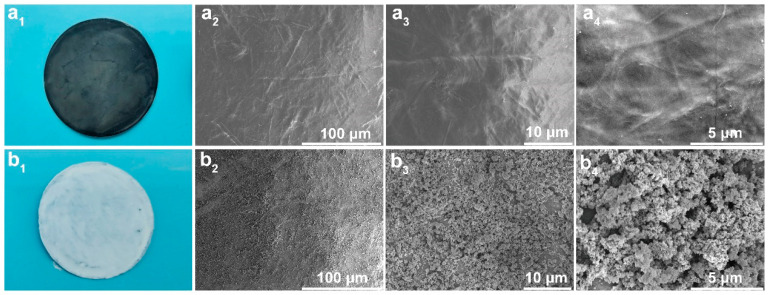
The digital photographs of L-CNT (**a_1_**) and L-CNT-TiO_2_ films (**b_1_**); surface morphologies of L-CNT (**a_2_**–**a_4_**) and L-CNT-TiO_2_ films (**b_2_***–***b_4_**).

**Figure 4 polymers-15-03666-f004:**
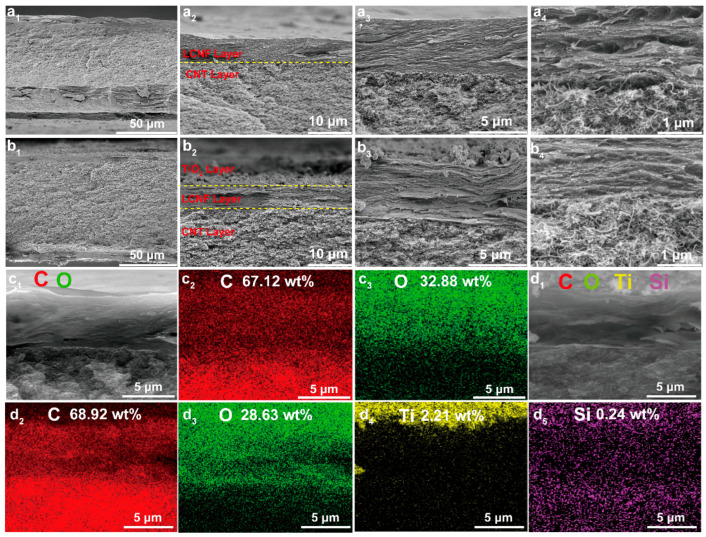
SEM images of the cross-sectional microstructures and the corresponding element mappings of L-CNT (**a_1_**–**a_4_**,**c_1_**–**c_3_**) and L-CNT-TiO_2_ (**b_1_**–**b_4_**,**d_1_**–**d_5_**) films.

**Figure 5 polymers-15-03666-f005:**
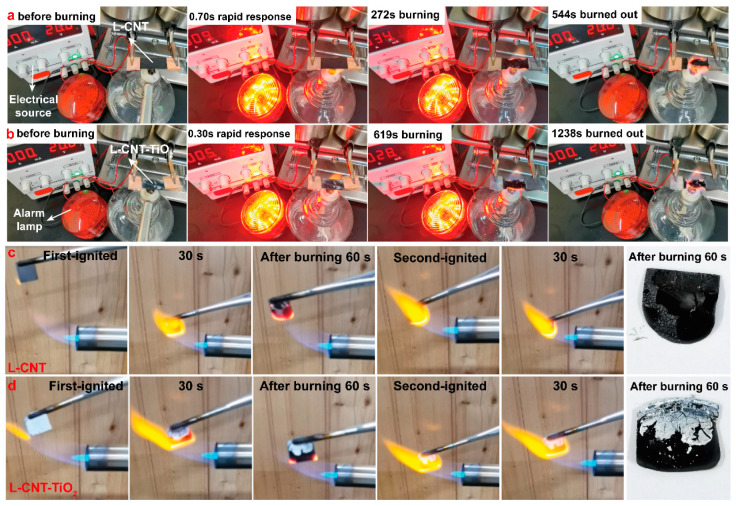
Snapshots of the entire fire-warning process of L-CNT (**a**) and L-CNT-TiO_2_ (**b**) films including trigger, alarm and signal disappearance; photographs of 120 s of the consecutive burning process of L-CNT (**c**) and L-CNT-TiO_2_ (**d**) films.

**Figure 6 polymers-15-03666-f006:**
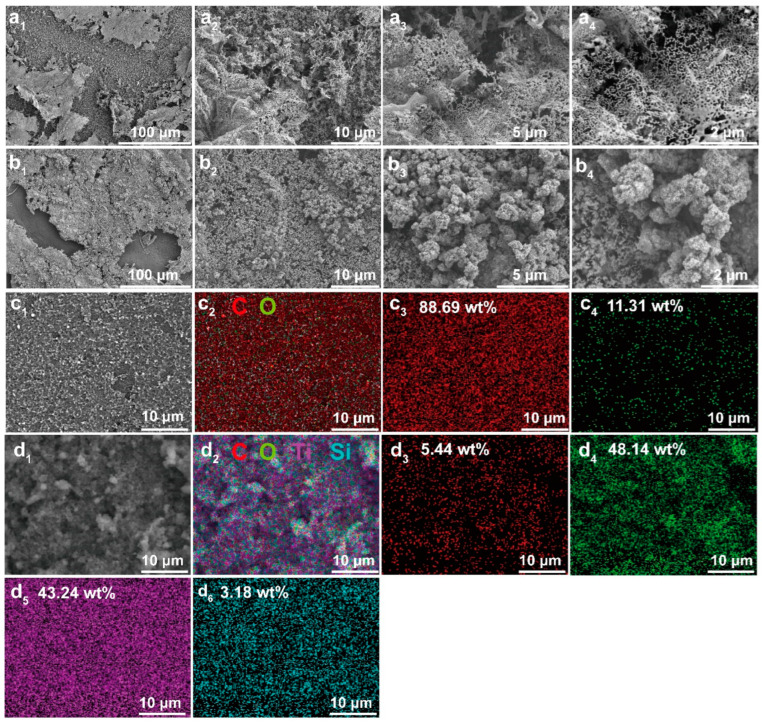
SEM images of the char microstructures and the corresponding element mappings of L-CNT (**a_1_**–**a_4_**,**c_1_**–**c_4_**) and L-CNT-TiO_2_ (**b_1_**–**b_4_**,**d_1_**–**d_6_**) films.

**Figure 7 polymers-15-03666-f007:**
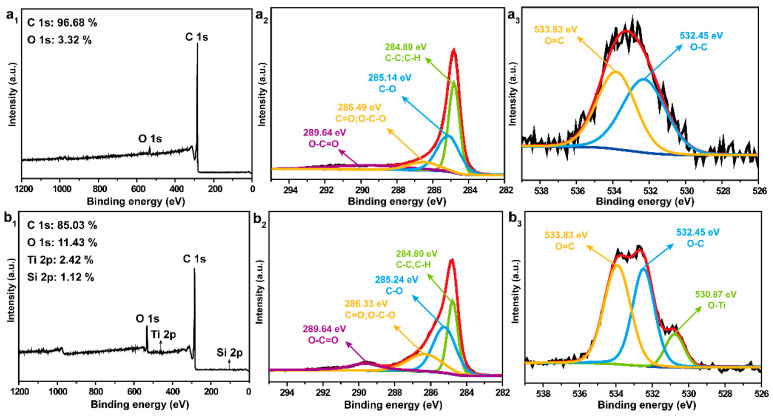
The XPS full spectra and the corresponding high resolution C 1 s, O 1s spectra of the char layers of L-CNT (**a_1_**–**a_3_**) and L-CNT-TiO_2_ (**b_1_**–**b_3_**) films.

**Figure 8 polymers-15-03666-f008:**
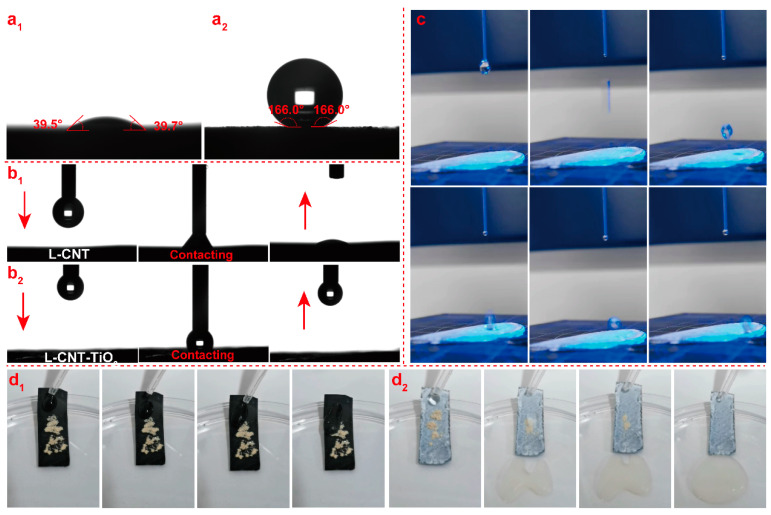
Photographs of water droplet contact angles on surfaces of the L-CNT (**a_1_**) and L-CNT-TiO_2_ (**a_2_**) films; anti-adhesion attraction between the water drops and L-CNT (**b_1_**) and L-CNT-TiO_2_ (**b_2_**) films; typical photographs of the perpendicular water droplet-impacting test for the surface of the L-CNT-TiO_2_ film (**c**); water droplets rolling off to remove dust from the surface of the L-CNT (**d_1_**) and L-CNT-TiO_2_ (**d_2_**) films.

**Table 1 polymers-15-03666-t001:** Comparison of early fire-warning performances between our CNT-based film and other systems in the literature.

Composition and Type of Composite Materials ^a^	Sensors	Alarm Trigger Time/s	Fire Duration Time/s	Ref.
FRPU@GO/CNTs@BN foam	GO/CNTs	~8	>120	[11]
CB@KF/CNT/PVA CF	CNTs	4	NM ^b^	[15]
CNTs@CCS/PANI@PAA	CNTs	~1	NM	[16]
GO/DCNT composite films	GO/CNTs	1	NM	[8]
PEG@wood/CNT/CA aerogel	CNTs	~2.03	NM	[31]
PAN/CNTs/APP fiber filters	CNTs	5	NM	[32]
Silver/Fe_3_O_4_ nanowire	Silver	2	>900	[33]
GO-PTA paper	GO	0.5	283	[34]
GO-PA paper	GO	~0.5	433	[35]
CNF-MMT-GO aerogel	GO	1.9	137	[36]
GO/TA/P-CNF paper	GO	1	>120	[37]
GO/HCPA nanocomposite papers	GO	0.6	>600	[38]
CNF-MMT-GO aerogel	GO	1.9	138.9	[36]
SPI/MSF/CA/GN film	GN	1	220	[39]
GO/silicone coating	GO	2~3	~90	[40]
SA/HAP/GO aerogel	GO	~1.5	>60	[41]
Phosphorylated GO/BNNS film	GO	<2	131	[42]
Graphene/SPI/MSF-g-COOH/CA/GN films	Graphene	1	>220	[39]
rGO@XNBR/MMT film	GO	<2	>180	[43]
LAA-MPMS-GO	GO	~1	>180	[3]
Chitosan/GO	GO	~3	1280	[44]
CNF-GO/APP agrogel	GO	2.6	129	[45]
BA/GO paper	GO	0.8	205.4	[46]
Polyimide/MXene aerogel	MXene	<5	>60	[47]
MXene/TA/CaCl_2_ cotton fabric	MXene	~3	NM	[48]
MXene/SMPU paper	MXene	4 s	NM	[49]
PVP/PEG-MXene networks	MXene	1.8	NM	[9]
PLCNF/gelatin/MXene aerogel	MXene	~1	NM	[50]
CAS/TA/MXene film	MXene	0.982	NM	[51]
CCS/MXene cotton fabric	MXene	3.8	NM	[52]
MXene/cellulose nanocoating	MXene	3.1	NM	[53]
PU/MXene paper	MXene	11	NM	[49]
L-CNT-TiO_2_ film	CNTs	~0.30	~1238	This work

^a^ Notes: FRPU: fire-retardant rigid polyurethane, BN: boron nitride, CB@KF: carbon black nanoparticles with core-shell structure, PVA: polyvinyl alcohol, CF: cotton fiber, FGO: flame-retardant modified-graphene oxide, PUS: polyurethane sponge, CCS: chitosan, PANI: polyaniline, PAA: polyacrylic acid, DCNT: dopamine-CNT, PEG: polyethylene glycol, CA: calcium alginate, PAN: polyacrylonitrile, APP: ammonium polyphosphate, PTA: phytic acid, PA: phosphoric acid, CNF: cellulose nanofiber, MMT: montmorillonite, TA: tannic acid, P-CNFs: phosphorylated-cellulose nanofibrils, HCPA: multi-amino molecule, SPI: soybean protein isolate, MSF: sisal cellulose microcrystals, CA: citric acid, GN: graphene, SA: sodium alginate, HAP: hydroxyapatite, PVP: polyvinyl pyrrolidone, BNNS: boron nitride nanosheets, SPI: soy protein isolate; MSF-g-COOH: sisal cellulose microcrystals; GN: graphene, CA: citric acid, XNBR: carboxylic acrylonitrile-butadiene rubber, LAA: L-ascorbic acid, MPMS: 3-methacryloxypropyltrimethoxysilane, BA: boric acid, SMPU: shape memory polyurethane, PLCNFs: phosphated lignocellulose nanofibrils, CAS: casein, PU: polyurethane. ^b^ NM: not mentioned.

## Data Availability

Not applicable.

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
