# Peer review of "Construction of Multifunctional Hierarchical Biofilms for Highly Sensitive and Weather-Resistant Fire Warning"

_polymers, 2023, doi:10.3390/polym15183666_

Round 1

Reviewer 1 Report

polymers-2588490

Construction of multifunctional hierarchical biofilms for highly-sensitive and weather-resistant fire warning

After careful evaluation, I have concluded that the above-mentioned manuscript requires major revision.

Comments

  1. Elaborate on the specific role of lignocellulose nanofibrils (LCNFs) in the hierarchical biofilm structure and their contribution to the fire response and flame resistance properties?
  2. How was the interaction between carbon nanotubes (CNTs) and LCNFs optimized to achieve the desired conductive and insulating characteristics for the fire-warning system?
  3. Explain how the electronic excitation characteristics of TiO2 contribute to the rapid transition from an insulating state to a conductive state during fire exposure?
  4. What were the challenges faced during the development of the multi-scale hierarchical biofilm? How were these challenges overcome?
  5. Regarding the observed micro-papillae and nano-scale protrusions on the surface of the L-CNT-TiO2 film, how do these features contribute to its superhydrophobic properties?
  6. Provide more details about the mechanism behind the improved flame resistance of the L-CNT-TiO2 film after TiO2-HDTMS modification? How does the silicon-containing cross-linked char layer interact with the TiO2 layer?
  7. What specific techniques were employed to characterize the thermal stability and fire-resistant properties of the TiO2 layer in the L-CNT-TiO2 film?
  8. In terms of the early fire-warning system, how did authors ensure the reliability and accuracy of the 0.30 s fire response signal? Were there any external factors that could potentially affect the system's performance?
  9. Discuss any potential limitations or challenges in scaling up the production of the L-CNT-TiO2 film for practical applications?
  10. Given the excellent UV protection factor (UPF) of the films, have authors considered potential applications in outdoor settings beyond fire safety, such as building facades or other structures exposed to sunlight?
  11. How versatile is the developed methodology for creating multifunctional films? Are there possibilities for incorporating additional functionalities beyond fire safety and UV protection?
  12. Elaborate on the process by which the hierarchical structure of the L-CNT-TiO2 film was characterized using SEM and EDS? How did this characterization inform your understanding of the film's properties?
  13. What challenges were encountered when assessing the mechanical robustness and durability of the L-CNT-TiO2 film, especially under harsh conditions or extended exposure?
  14. Discuss the potential impact of environmental factors, such as humidity and temperature variations, on the long-term performance of the L-CNT-TiO2 film's superhydrophobicity and fire-warning capabilities?
  15. Provide more information on the potential cost-effectiveness of producing the L-CNT-TiO2 film at scale, considering its multiple functionalities?
  16. How might the presence of Ti and Si elements affect the overall environmental impact of the L-CNT-TiO2 film, especially considering its potential applications in fire safety and building materials?
  17. Are there any plans to optimize the synthesis process to further enhance specific properties, such as fire response time, flame resistance, or UV protection, without compromising others?
  18. Given the complex interplay of materials and functionalities, have authors considered the potential trade-offs between different properties (e.g., fire resistance vs. UV protection) and how to optimize them for specific applications?
  19. How does the performance of your L-CNT-TiO2 film compare with existing fire alarm and flame resistance technologies in terms of sensitivity, response time, and practicality?
  20. Provide insights into the potential regulatory challenges or certifications that might be required for the adoption of the L-CNT-TiO2 film in fire safety applications?
  21. Considering the hydrophobic nature of the film, how might the self-cleaning and anti-fouling properties be affected by long-term exposure to environmental pollutants and contaminants?
  22. Discuss any ongoing research or future plans to expand the capabilities of the L-CNT-TiO2 film, such as integrating sensors for real-time monitoring or incorporating additional functionalities?
  23. Given the reliance on TiO2, a relatively expensive material, have authors explored strategies to potentially reduce the material cost without compromising the film's performance?
  24. What challenges do authors foresee when transitioning from laboratory-scale production to large-scale manufacturing of the L-CNT-TiO2 film, especially in terms of maintaining consistent performance and quality?
  25. How might the material composition and structure of the L-CNT-TiO2 film affect its potential recyclability or end-of-life disposal considerations?
  26. In terms of the fire-warning system's sensitivity and reliability, have authors conducted any tests with variations in fire intensity, duration, or different types of fire sources to validate its performance under various scenarios?
  27. Elaborate on the potential impact of the hierarchical structure and material composition of the film on its mechanical properties, such as tensile strength and flexibility?
  28. What insights could authors provide regarding the potential limitations of the L-CNT-TiO2 film, such as its performance in extreme weather conditions or its compatibility with different building materials?

Minor editing of English language required.

Reviewer 2 Report

This article dealt with the fire alarm performances and surface properties of three-layered CNT, LNCF, and TiO2 films. The subject of fire alarm system showing fast response time and long-term working time is important and deserves attention. It may be publishable after clarifying and modifying following issues.

1. The device consists of CNT, LNCF, and TiO2-HDTMS layers. What is the thickness of each layer and its effect on the fire alarm performances? Is there optimum thickness of each layer in terms of fire performances?

2. For practical uses, the device should be stable even in an outdoor environment as mentioned in the introductory part. Delamination often occurs in a LBL film when exposed to high/low temperature and humid due to weak interaction between layers and difference in intrinsic properties of materials. Are there any problems related to delamination and long-term stability?

3. Please define ‘FAWs’ and ‘UPF’ in a revised manuscript for general readers.

There are many typos and grammatical errors throughout the manuscript. For example, ‘For detecte’ should change to ‘for detecting’ in the introductory part and ‘Uv-resistance’ should be ‘UV-resistance’ in the experimental part. Some grammatical errors are also found in the manuscript. Please read the manuscript carefully and revise the typos and grammatical errors.

Round 2

Reviewer 1 Report

The revised manuscript can be accepted